# Cooperative spin crossover leading to bistable and multi-inert system states in an iron(III) complex

Andreas Dürrmann [1,2], Gerald Hörner [1,2], Dirk Baabe[3], Frank W. Heinemann [4], Mauricio A. C. de Melo[5] & Birgit Weber [1,2] ✉

Cooperativity among spin centres has long been the royal road in spin crossover (SCO) research to impose magnetic bistability in terms of thermal hysteresis. In this work we access magnetic multi-inert states of the iron(III) compound {FeL$_2$[B(Ph)$_4$]} ≡ FeB at low temperature, in addition to thermal bistability. The packing of the low-spin and high-spin forms of crystalline FeB differs only marginally what ultimately leads to structural conservatism. This indicates that the SCO-immanent breathing of the complex cation is almost fully compensated by the anion matrix. The unique cooling rate dependence of the residual low-temperature magnetisation in FeB unveils continuous switching between the trapped high-spin (ON) and the relaxed low-spin state (OFF). The macroscopic ratio of the spin states (ON:OFF) can be adjusted as a simple function of the cooling rate. That is, cooperative spin crossover can be the source of bistable and multi-inert system states in the very same material.

Controlled, clear-cut shuttling of a system between the system states A and B (and C, D, etc.) is the basic requirement of a (macroscopic) switch. A macroscopic switch with two different states provides direct ON/OFF feedback for the desired application. When switching as a concept is to be transposed to molecular dimensions, however, the need for bistability (or multistability) of the miniaturized system conflicts with very fundamental laws of thermodynamics. That is, either A or B represents the stable state of the system, while the ratio A/B is a continuous and single-valued function of temperature, governed by the enthalpy and entropy of the system (i.e., the Boltzmann-like analog-signal case in Fig. 1a). Solutions to this intricate molecular problem are offered by cooperative concatenation. Strong cooperativity renders the shuttle between discrete system states A and B abrupt and discontinuous (i.e., the digital 0/1 case in Fig. 1b), whereas favourable instances indeed allow for bistability: For such cases, the system status is no longer defined solely by molecular thermodynamics but reflects the history of the bulk system (i.e., the loop-like case in Fig. 1c). Therefore,

the switching between (bi)stable system states A and B of molecular materials relies entirely on the molecular ensemble, which provides the active playground for hysteresis. The obvious analogy to silicon-based microelectronics has motivated intense search for hysteresis behaviour in molecular materials.

Prototypical real-world examples for the state-plots in Fig. 1a–d are provided by the temperature-dependent magnetic response of metal complexes which undergo spin crossover (SCO) transitions[1,2]. The phenomenon of SCO is a well-investigated example for molecular switchability in materials based on 3d metal centres[3,4]. The metal ion can attain two boundary spin configurations, the high-spin (HS, 1/ON) and the low-spin state (LS, 0/OFF), depending on external compulsions (e.g., Δ$T$, Δ$p$, $hv$). The microscopic SCO transition is coupled to variation of macroscopic properties so that the macroscopic system state can be followed quantitatively (in terms of the HS fraction $\gamma_{HS}$), supported by a magnetometer or spectroscopic methods[5,6]. For almost 30 years, research on thermal SCO has elaborated on the theme of gradual, abrupt or bistable spin transitions (Fig. 1a–c)[7], through synthetic

[1]Institute for Inorganic and Analytical Chemistry, Friedrich Schiller University Jena, Humboldtstraße 8, Jena, Germany. [2]Inorganic Chemistry IV, University of Bayreuth, Universitätsstraße 30, Bayreuth, Germany. [3]Institut für Anorganische und Analytische Chemie, Technische Universität Braunschweig, Hagenring 30, Braunschweig, Germany. [4]Lehrstuhl für Anorganische und Allgemeine Chemie, Friedrich-Alexander-Universität Erlangen-Nürnberg, Egerlandstr. 1, Erlangen, Germany. [5]Departamento de Fisica, Universidade Estadual de Maringá, Maringá, Brazil. ✉e-mail: birgit.weber@uni-jena.de

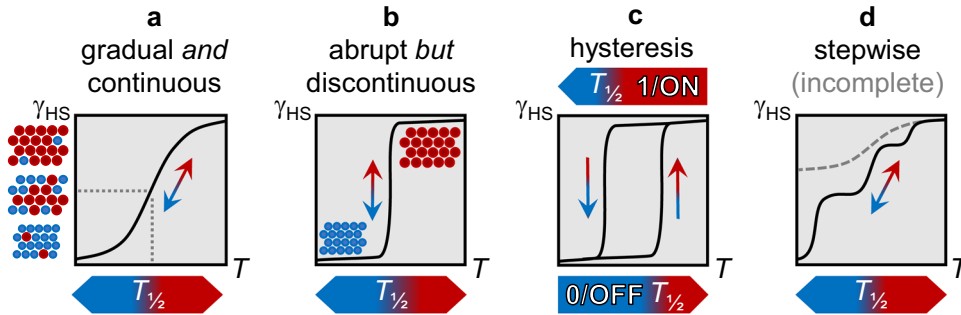

**Fig. 1 | Established SCO types of molecular ensembles, plotted as $\gamma_{HS}$ vs. $T$;** $\gamma_{HS}(T_{1/2}) = 0.5$. **a** Analog Boltzmann-like distribution observed for less cooperative systems or in solution. **b**, **c** Case of strong cooperativity for digital bistable system states. **d** Stepwise (or incomplete) transition; can also occur with hysteresis. The slope of the single steps may differ. Coloured circles represent the HS (red) or LS (blue) state, respectively.

approaches and crystal engineering[8]. For a brief account on the theoretical background on SCO, please refer to Supplementary Note 1.

Various concepts were introduced to interpret the different transition types and to relate them to (supra)molecular parameters[9–11]. Nevertheless, the restriction to binary system states invariantly limits the utility of SCO systems: Irrespective of the way how the HS↔LS transition proceeds, in the end the HS ($\gamma_{HS}$ = 1/ON) or the LS state ($\gamma_{HS}$ = 0/OFF) is observed at a given temperature. Spin transitions running over more than only one single step indicate a way towards multistability (Fig. 1d)[12–14]. However, concepts are sparse which would address three different states, A, B, and C[15], while the number of examples is even more limited when we go beyond a three-state system.

Only a few iron(II) complexes were reported, where the thermal transition temperature $T_{1/2}$ coincides with the existing curve of metastable states (temperature- or light-induced excited spin state trapping; TIESST or LIESST). Such intersecting phenomena have been rationalised in detail by L'Étard et al.[16]. The convolution of the kinetic and thermally activated domain can result in, e.g., oligo-metastability below a threshold temperature $T_{crit}$. In these cases, the switching remained again incomplete, even at low temperatures, so that only a limited fraction of the potential system states could be accessed[17,18]. A final example was reported for a valence-tautomeric cobalt complex with a scan rate dependent hysteresis width and HS fraction, which once more suffered from very incomplete spin transitions[19]. In general there is yet a lack of accessing such oligo-metastable states from a smooth, unidirectional progression since known systems rely on subtle details of the experimental conditions[20,21]. As it is noted that the common term 'metastable' rather floppily mixes concepts from thermodynamics and kinetics, in the remainder of this article we refer to 'multi-inert' states.

Just as stated by other authors, in-depth analyses of SCO systems with thermal hysteresis are a base requirement of a more profound understanding of the SCO process in the solid state[22]. Herein, we report such a highly detailed SQUID magnetometry study, complemented by $^{57}$Fe Mössbauer spectroscopy and $T$-dependent single crystal diffraction for the complex salt {FeL$_2$[B(Ph)$_4$]} ≡ FeB with *NN'O* bis-meridional coordinated iron(III). As becomes evident, a unique combination of thermodynamic and kinetic factors arises from structurally conservative SCO in the crystal lattice of compound FeB. It allows complementary access to digital system states ($\gamma_{HS}$ = 0 and $\gamma_{HS}$ = 1) with thermal hysteresis on the one hand and to TIESST-related analog states on the other hand. Besides the non-exceptional black-or-white switching, we noticed that (kinetically) multi-inert system states can be freely selected at $T_{crit} < T_{TIESST}$: The final system state at $T \leq T_{crit}$ is therefore not limited to black-or-white (high- or low-spin only) but allows for all shades of grey (i.e., all ratios between high- and low-spin), rendering the material bistable and multi-inert at the same time. We

feel that the potential of compound FeB and related materials for multi-state switching, which has been previously neglected, cannot be overemphasized.

## Results
### Preparation

The monobasic, tridentate *NN'O* proto-ligand HL was prepared in three steps from imidazole as starting material in total yields up to 23% (Fig. 2a, b). The conversion of anhydrous ferric chloride with two equivalents of the meridional-coordinating ligand HL gave a wine-red suspension from which compound FeB could be precipitated as a dark red crystalline solid by metathesis with Na[B(Ph)$_4$]. It is noted that the entire crop consisted of high-quality single crystals (Fig. 2c). High-resolution mass spectrometry, elemental combustion analysis, and single crystal X-ray diffraction (*vide infra*) confirm the stoichiometric composition of compound FeB.

### Crystallography

Three observations in the X-ray diffraction data of compound FeB, collected at $T$ = 120 and 65 K, are discussed in detail (Fig. 3 and Table 1; complete numbering and crystallographic details are listed in Supplementary Table 2 and Supplementary Fig. 6–8).

(*i*) The molecular metrics at low and high temperature are in full agreement with the low-spin and the high-spin state, respectively. (*ii*) Orthorhombic crystal symmetry of space group *Pbcn* is conserved at both temperatures and, what is more, the lattice volume and the underlying vectors exhibit marginal variation with temperature. SCO is crystallographically conservative since the SCO-related breathing of the complex entity (FeL$_2$)$^+$ is compensated by the bulky [B(Ph)$_4$]$^-$ anions. (*iii*) A 2D-network of >N$_{Im}$H···O=C< hydrogen bonds between neighbouring (FeL$_2$)$^+$ units within the *ac* plane strengthens and weakens at low and high temperature, respectively.

Massively shortened bonds (av. Fe–N contracts from 2.08 to 1.92 Å; Fe–O from 1.95 to 1.89 Å; cf. Figure 3c) and significantly reduced angular distortion (reduction from $\Sigma$ = 80.9° to 43.0° and from $\Theta$ = 309.4° to 134.3°) accompany the SCO from the HS state at $T$ = 120 K to the LS state at $T$ = 65 K[23]. An overlay of the corresponding structures is given in Fig. 3b and selected bond lengths and angles are summarized in Table 1. Taken together, intra-complex torsion and non-linear breathing of the coordination sphere displace non-H atoms at the ligand periphery by up to 0.7 Å (see structure overlay in Supplementary Fig. 16)[24]. This motion indeed proves to be relevant for the strength of the H-bonding of adjacent complex units (*vide infra*). Finally, also the dihedral angle θ defined by the two ligand planes is susceptible towards a spin state change due to twisting of the meridional ligands ($\Delta\theta^{HS–LS}$ = 1.8°; cf. Supplementary Fig. 17)[25]. According to Halcrow's observation for $\Delta\theta^{HS–LS}$ > 1°, this finding suggests that kinetic effects will accompany low-temperature SCO[26,27]. Indeed, SCO in

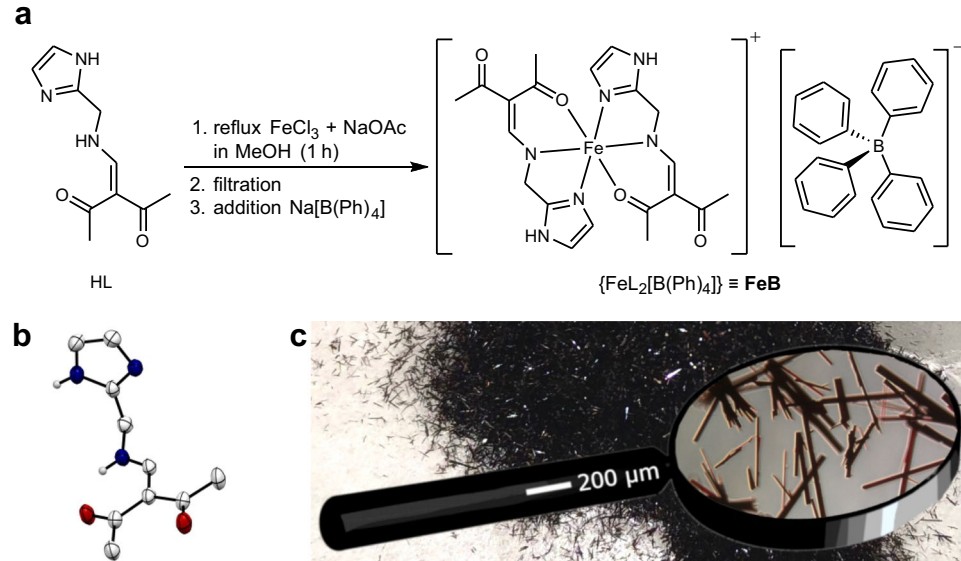

**Fig. 2 | Chemical aspects of compound FeB. a** Synthesis of compound FeB. The complete synthetic route including the ligand HL is given in Supplementary Fig. 2. **b** Molecular structure of HL obtained from single crystal data (Supplementary Table 1 and Supplementary Figs. 4 and 5). Displacement ellipsoids are drawn at 50% probability level. Hydrogen atoms connected to carbon atoms (spheres in arbitrary size) are omitted for clarity. **c** Picture of single-crystalline FeB (see also Supplementary Fig. 3).

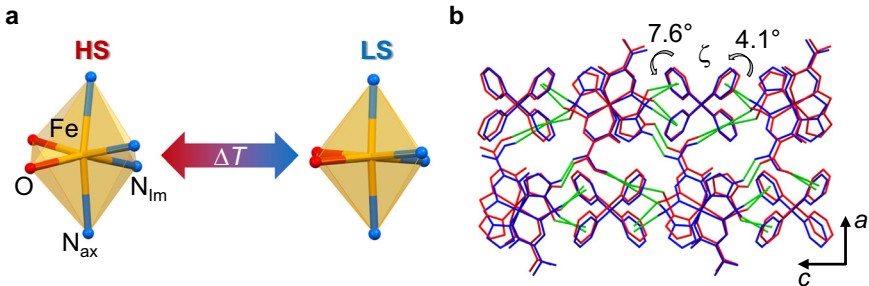

**Fig. 3 | Structural aspects of compound FeB. a** Inner coordination sphere of $(FeL_2)^+$ in the HS and LS state. **b** Overlay of the crystal packing in the *ac* plane of the HS (red) and LS (blue) structures with the three dominant intermolecular contacts ($N_{Im}H\cdots O$, $CH_2\cdots\pi$, $C_{Me}H\cdots\pi$), drawn in dashed green lines. The angle $\zeta$ denotes torsional rotation of phenyl planes ('anion displacement') upon spin transition, please see section Discussion. Hydrogen atoms not involved in H-bonding are omitted for clarity. More illustrations as well as Hirshfeld surface analysis are provided in Supplementary Fig. 9–15.

**Table 1 | Selected metrics of the solid-state structure**

|  | Space group | $d(Fe–N_{Im})$ / Å | $d(Fe–N_{ax})$ / Å | $d(Fe–O)$ / Å | $\angle(N_{Im}–Fe–O)$ / ° | $d(N_{Im}H\cdots O)$ / Å | $\angle(N_{Im}H\cdots O)$ / ° | $V_{cell}$ / Å³ |
|---|---|---|---|---|---|---|---|---|
| HS | *Pbcn* | 2.075 | 2.096 | 1.953 | 161.09 | 2.35 | 120 | 3885.9 |
| LS | *Pbcn* | 1.937 | 1.913 | 1.894 | 172.16 | 2.01 | 139 | 3842.0 |
| \|Δ\| | n.a. | 0.138 (6.6%) | 0.183 (8.7%) | 0.059 (3.0%) | 11.07 (6.4%) | 0.34 (14.5%) | 19 (13.7%) | 43.9 (1.1%) |

Detailed listing of metrics is provided in Supplementary Table 3–6. $N_{Im}$ and $N_{ax}$ denotes the imidazole nitrogen or axially coordinating nitrogen atom, respectively (see also Fig. 3a).

compound FeB is found to be governed by kinetic effects (see following sections).

SCO-related build-up of local strain in the crystal may be amplified to enforce global consequences such as cooperativity. In many cases, and very much so in cooperative systems, this strain culminates in crystallographic phase transitions with discrete, discontinuous changes in lattice parameters. Quite unusually in crystalline FeB, the associated expansion and contraction of the cation during spin state switch neither translates into changes of crystal symmetry nor into significant variation of the cell dimensions. The shortening of axis *b* and *c* by 2.9% and 2.5% is almost completely compensated by elongation of axis *a* by 4.2% what leaves the cell volume conserved ($|\Delta V| = 44$ Å³; $|\Delta V| / V \approx 1\%$, see Table 1). While macroscopic anisotropy may drive the effect of 'jumping crystals'[28] in some symmetry-conserved SCO materials, the strain is quenched autogenously in compound FeB. More importantly, the orthorhombic space group *Pbcn* persists at both temperatures ($T = 120$ and 65 K). Consequently, we find the powder X-ray diffraction pattern at $T = 140$ K (HS) and 15 K (LS) almost unchanged (Supplementary Fig. 18 and 19). Although the observation of a strongly cooperative spin transition without first order phase transition is no precedential case[29], the origin in this particular case might be (see section Discussion below).

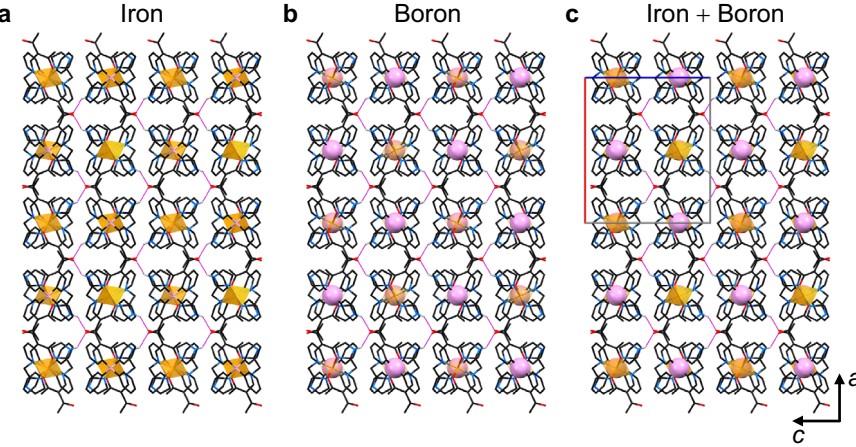

**Fig. 4 | Excerpt of the solid-state structure of compound FeB at $T$ = 65 K (LS) within the $ac$ plane (projection along axis $b$). a, b** Spatial alignment of the iron and boron sublattices, highlighted as sphere (B) or octahedron (Fe), respectively. **c** Visualisation of the chequered pattern and the unit cell (rectangle). The relevant H-bond $>N_{Im}H···O=C<$ is highlighted in magenta lines. Organic backbone of ligand and counterion are given as wireframe. Hydrogen atoms not involved in H-bonding are omitted for clarity.

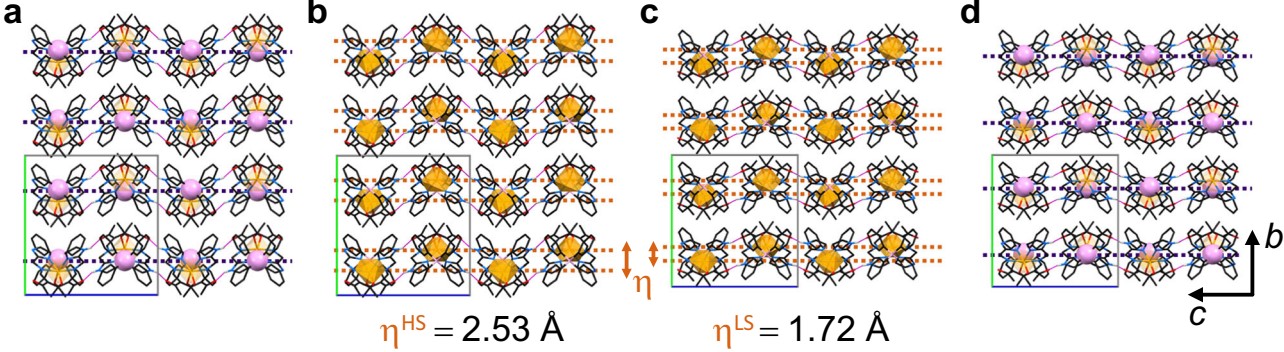

**Fig. 5 | Excerpt of the solid-state structure of FeB in the HS (a,b) and LS (c,d) state within the $bc$ plane (projection along axis $a$). a, d** View onto planar $[B(Ph)_4]^-$ layers. The purple dotted lines are guidance for the eye to underline the coplanar array of $[B(Ph)_4]^-$ anions. **b, c** Illustration of the H-bond directed 'Fe–Fe zig-zag wave'. The orange dotted lines are guidance for the eye to estimate the wave amplitude η. The relevant H-bond $>N_{Im}H···O=C<$ is highlighted in magenta lines. Iron and boron sublattices are highlighted as sphere (B) or octahedron (Fe), respectively. Organic backbone of ligand and counterion are given as wireframe. Hydrogen atoms not involved in H-bonding are omitted for clarity. The unit cell is drawn as a rectangle.

Figure 4 captures the highly symmetric packing of anions and cations in the $ac$ plane where a close-to-ideal chequered pattern emerges as a distorted variation of the rock salt lattice. Here, cation and anion fill the space equally due to appropriate spatial requirements of both building blocks, so that there is no solvent accessible void in those 2D planes. Similar structures have been observed in other iron(III) compounds with $[B(Ph)_4]^-$ as counterion, but such a chequerboard is *not* necessarily the result of this combination[30]. Overall, both strong and directional intermolecular interactions shape the chequered pattern in the $ac$ layer, in particular the combination of hydrogen bonds between individual complex molecules ($>N_{Im}H···O=C<$) and CH···π contacts between $(FeL_2)^+$ and $[B(Ph)_4]^-$. It is noted that the complementary packing of individual 2D layers along $b$ lacks any strong interactions. That is, the 3D supramolecular structure solely relies on weak non-classical $C_{Me}H···π$ contacts in the third cell dimension.

Intriguingly, none of the above qualitative and quantitative structural features within the crystal undergo significant changes when the complex molecules switch between the HS and LS states. We believe that the conservative character of compound FeB is the very reason for the resilience of the samples against iterative thermal agitation during the cooling-heating cycles (*vide infra*).

The notion of structural conservatism extends to the non-bonded Fe···Fe distance in the $ac$ plane, which stays virtually unchanged (11.39 Å in the HS form to 11.38 Å in the LS form). While the SCO centres keep their distance constant, their relative orientation is subject to significant variation with the spin state. The almost ideal square planar anionic sublattice of borate centres in the $ac$ plane is interconnected through short H···H contacts of adjacent phenyl moieties with $d$(H···H) ≈ 2.2 Å. Even shorter contacts prevail for the constituents of the cationic $(FeL_2)^+$ sublattice. H-bonding between ligand-appended $>N_{Im}H$ and $O=C<$ groups employ each of the H-bond donors and acceptors of $(FeL_2)^+$ to form an infinite 2D-network of H-bonds within the $ac$ plane (Fig. 4).

The directionality of the H-bonds displaces the iron centres, and no coplanar arrangement is observed. In other words, the non-coplanar network is a zigzag wave above and below the borate layer (Fig. 5a, d), which is derived from complex units tilted by 180°. Notably, the amplitude of the wave (η) is significantly larger in the HS case than for the LS ($η^{HS}$ = 2.53 Å and $η^{LS}$ = 1.72 Å; cf. Fig. 5b, c).

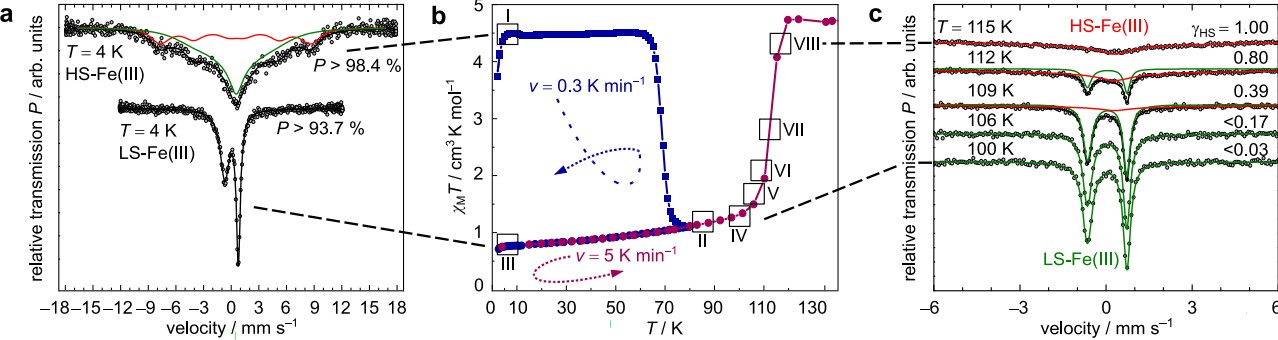

**Fig. 6 | Magnetic profile, SCO relaxation data and selected $^{57}$Fe Mössbauer spectra of compound FeB. a** Zero-field $^{57}$Fe Mössbauer spectra at $T = 4$ K of the trapped $HS_{TR}$ state (top) after quench-cooling ($v_\downarrow > 10$ K min$^{-1}$) and the relaxed LS state (bottom) after a cooling and heating sequence (I → II → III: $T = 4 → 90 → 4$ K). Symbols represent experimental data, solid lines are derived from a fit with the Blume-Tjon relaxation model. **b** TIESST relaxation experiment as a plot of $\chi_M T$ vs. $T$. Fast cooling from room temperature (I: $T = 300 → 4$ K; $v_\downarrow = 10$ K min$^{-1}$) was followed by progressive heating (I → II: $T = 4 → 90$ K; $v_\uparrow = 0.3$ K min$^{-1}$) and reverse cooling (II → III: $T = 90 → 4$ K; $v_\downarrow = 5$ K min$^{-1}$). Solid lines serve as a guidance for the eye. **c** Zero-field $^{57}$Fe Mössbauer spectra recorded at $T = 100$, 106, 109, 112 and 115 K after quench-cooling and subsequent warming (I → II → III → IV → VIII: $T = 4 → 90 → 4 → 100 → 115$ K). Symbols represent experimental data, solid lines are derived from a fit with the Blume-Tjon relaxation model. Source data are provided as a Source Data file.

The overall flattening of the iron sublattice within layer *ac* goes along with a significant shortening of the H-bonds from 2.350 Å in the HS to only 2.012 Å in the LS state. Concomitant with the contraction, we see a substantial linearisation of the H-bond; the angle ∡(N−H⋯O) widens from 119.5° to 139.1° in the HS and the LS state, respectively. Taking these factors together, the H-bond network undergoes a transition from 'weak' to 'moderate'[31]. Given the importance of H-bonding for cooperativity[32], we must expect that this sharp variation in the H-bond shape and strength will feedback to the SCO properties. Indeed, SCO in compound FeB is governed by strong cooperativity which reflects in the parallel observation of bistable regimes and multi-inert states.

## Bistability, Thermal Trapping and Kinetic Multi-inert States

As an orienting magnetometry experiment with the default nominal temperature scan rate of $v = 5$ K min$^{-1}$ indicated dynamic processes on the SQUID timescale, we devised a comprehensive study of the scan rate response on independently synthesised samples of compound FeB. For the sake of clarity, we will present our findings as an extract of the whole data set. Please refer to Supplementary Table. 7–9 and Supplementary Fig. 20–42 to find the complete information about magnetic susceptibility recordings. In addition, we used the complementary $^{57}$Fe Mössbauer spectroscopy to further characterise the SCO in FeB; this technique is well-suited for probing the temperature-dependent development of (co)existing LS and HS states. As quantitative analyses of $^{57}$Fe Mössbauer spectroscopy in recent iron(III) SCO research are scarce[33,34], we like to take the chance to discuss the observations in some detail. First, both methods fully deliver the same picture:

Quench-cooling from $T = 300$ K to 4 K yields a trapped $HS_{TR}$ state ($\gamma_{HS} = 1$; I in Fig. 6b), while the relaxed LS state ($\gamma_{HS} = 0$) is detected by (zero-field) $^{57}$Fe Mössbauer spectroscopy after sequential heating to $T = 90$ K and reverse cooling to $T = 4$ K (II and III in Fig. 6b). With SQUID magnetometry, the abrupt relaxation of the trapped $HS_{TR}$ state to the thermodynamic stable LS state is consistently observed at $T_{TIESST} \approx 67$ K (temperature-induced excited spin state trapping; I → II in Fig. 6b)[16,35,36], which is an exciting result, given the vanishing number of previous reports in iron(III) chemistry[37–40]; in fact, there is just a single example with a higher $T_{TIESST}$ of 75 K[37]. That is, we see thermal switching between $\gamma_{HS} = 1$ and $\gamma_{HS} = 0$ in a binary regime ($HS_{TR}→LS$ transition) which unfolds a 'hidden' thermal hysteresis of 43 K in the range between $T_{TIESST} = 67$ K and $T_{1/2\uparrow} = 110$ K. With the Mössbauer signatures of HS and LS states of FeB being known, the temperature-dependent spin state ratio was resolved along the LS→HS transition

upon sequential warming of the sample (second binary regime; Supplementary Table 10 and Supplementary Fig. 43). A complete conversion is gradually captured between $T = 100$–115 K with $\gamma_{HS}(100$ K$) < 0.03$ increasing to $\gamma_{HS}(115$ K$) = 1.00$ (IV → VIII in Fig. 6c). Descriptive explanations concerning Mössbauer spectroscopy are provided in Supplementary Tables 10–12 and Supplementary Fig. 1, 43–50.

The value of the magnetic susceptibility-temperature product ($\chi_M T$) for the relaxed LS state equals 0.73 cm$^3$ K mol$^{-1}$ at $T = 2$ K, which indicates a system state of $\gamma_{HS} = 0$. Slight deviations from the spin-only limit (with $\chi_M T = 0.375$ cm$^3$ K mol$^{-1}$) echo reports by other authors for ferric ions with $S = \frac{1}{2}$[41]. Crystal lattice defects may be a source of deviation[42], whereas significant volume fractions of residual HS species can be ruled out from $^{57}$Fe Mössbauer spectroscopy. Here, a strongly asymmetric but well-resolved doublet is observed in the zero-field spectrum at $T = 4$ K (Fig. 6a). Such an asymmetric (and temperature dependent; cf. Figure 6c) line shape of an individual quadrupole doublet of a polycrystalline powder sample is indicating the presence of paramagnetic relaxation that is slow compared to the Larmor precession time of the $^{57}$Fe nuclear magnetic moment[43,44]. Therefore, to also account for such relaxation processes, a single-site analysis on basis of the stochastic Blume-Tjon relaxation model[45] was used to determine the Mössbauer parameters, yielding an isomer shift of $\delta = 0.16$ mm s$^{-1}$ and a large quadrupole splitting of $|\Delta E_Q| = 1.47$ mm s$^{-1}$ (Supplementary Table 12). These values are in good agreement with literature reported values for an iron(III) LS state in an octahedral ligand field[46–48]. Supplementary measurements with an applied magnetic field ($H_{ext} = 5$ kOe) revealed a spectrum for the relaxed LS state with two components (subspectra) which give unexceptional Mössbauer parameters at $T = 4$ K (Supplementary Table 11 and Supplementary Fig. 44). The main difference between these two components are different fluctuation rates of the magnetic hyperfine field. Hence, since zero-field splitting is not possible in case of an $S = \frac{1}{2}$ spin state, we can safely attribute the observation of a two-component spectrum to packing effects in the solid state, which could cause small variations of the local microstructure around an individual iron nucleus, resulting in a (bimodal) distribution of relaxation times.

Upon heating to $T_{1/2\uparrow} \approx 110$ K, the steep jump of $\chi_M T$ in the magnetic profile to values around 4.6 cm$^3$ K mol$^{-1}$ is in accordance with an iron(III) centre in the HS state[23,46–48]. Full thermal conversion to the HS state ($\gamma_{HS} = 1$) is also observed in zero-field Mössbauer spectra at $T > 115$ K. The corresponding spectrum of FeB at room temperature exhibits an asymmetric and significantly broadened absorption line that can be fitted to parameters of a single $^{57}$Fe site (Supplementary

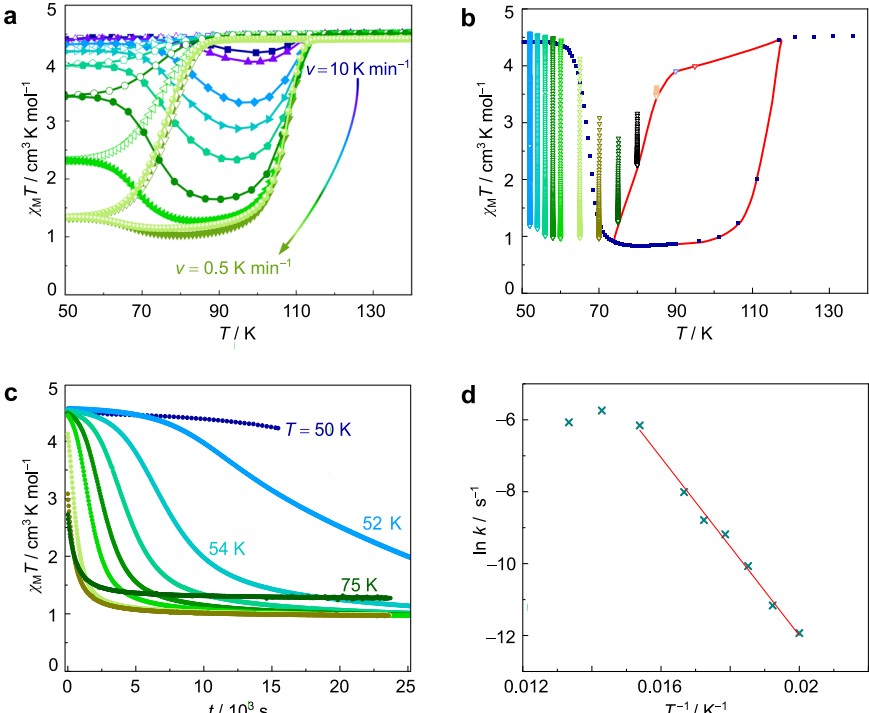

**Fig. 7 | Scan rate dependency and isothermal relaxation behaviour of compound FeB. a** Response on differing cooling/heating rates $v$, plotted as $\chi_M T$ vs. $T$ ($v$ = 10, 8, 5, 4, 3, 2, 1, 0.5 K min$^{-1}$, and settle mode). Settle mode is drawn in light green spheres. Open symbols represent cooling mode ($v_\downarrow$), filled symbols represent heating mode ($v_\uparrow$). Solid lines are drawn for clarity and do not represent data fittings or simulations. **b** Isothermal relaxation data, TIESST relaxation experiment and derived quasi-static thermal hysteresis loop (drawn in red). Vertically running data originate from relaxation studies. **c** Time-dependent evolution of $HS_{TR}\rightarrow LS$ relaxation at constant temperatures ($T$ = 50, 52, 54, 56, 58, 60, 65, 70, 75 K). **d** Plot of ln $k$ vs. $T^{-1}$. Red line represents Arrhenius fit. Details are provided in Supplementary Table 9 and Supplementary Fig. 42. Source data are provided as a Source Data file.

Fig. 45). The line-broadening at $T$ = 300 K and the increasing linewidth with decreasing temperature again indicate the presence of slow paramagnetic relaxation, also for the pure HS state of FeB at elevated temperatures (Supplementary Fig. 46 and 47). An analysis with the Blume-Tjon relaxation model[45] reveals isomer shifts in the range between δ = 0.31 and 0.55 mm s$^{-1}$, and only a small quadrupole splitting of $|\Delta E_Q|$ = 0.26 mm s$^{-1}$ was found at $T$ = 300, 200 and 150 K (Supplementary Table 10 and 12). Both numerical findings are in line with those observed for related iron(III) HS spin complexes[46–48]. In particular, the small values of $\Delta E_Q$ suggest only minor contributions of the surrounding ligand anions or valence electrons to the local electric field gradient, as it is typically expected for an iron(III) HS state with zero orbital momentum and a fairly octahedral coordination of the metal centre[44,49].

The LS→HS transition shows only minor dependence on the heating rate $v_\uparrow$ ($T_{1/2\uparrow}$ ≈ 110 K). By contrast, the system state strongly responds to variation of $v_\downarrow$ in the cooling runs, giving rise to an exceptional scan rate dependent branching for temperatures $T$ < 130 K as shown in Fig. 7a ($T_{1/2\downarrow}$ ≈ 75 K). The lowest values of $\chi_M T$ (1.0–1.34 cm$^3$ K mol$^{-1}$) were recorded for $v_\downarrow$ < 0.5 K min$^{-1}$ or in the settle mode (average $v_\downarrow^{settle}$ = 0.3 K min$^{-1}$), which means that even the temperature equilibration of settling does not allow for full HS to LS conversion[17].

The thermal hysteresis width in the settle mode of 31 K does also not vary critically compared to the hysteresis gaps obtained by sweep measurements (38–30 K; $\Delta T_{1/2\downarrow}$ = 4 K and $\Delta T_{1/2\uparrow}$ = 3 K for $v$ = 10–0.5 K min$^{-1}$; cf. Supplementary Table 8). The indifferent hysteresis width is quite significant an observation if compared to other scan rate dependent SCO complexes where $\Delta T_{1/2\downarrow}$ alone varies by up to 18.5 K[50]. Even when kinetic artifacts are entirely suppressed through isothermal decay to extract the pure thermodynamic SCO regime, there remains a quasi-static thermal hysteresis loop[16,17,35] of ca. 30 K (Fig. 7b, drawn in red). This is a sign of true bistability in the temperature range $T$ = 75–110 K.

Bringing the sample down to $T$ ≤ 50 K at large rates $v_\downarrow$ was found to leave $\chi_M T$ unchanged. That is, the HS state is trapped completely, giving a long-lived $HS_{TR}$ species at $T$ ≤ 50 K ($\chi_M T$ = 4.4 cm$^3$ K mol$^{-1}$; the dynamics of formation and decay of the trapped $HS_{TR}$ are discussed in detail below). This assignment is again supported by $^{57}$Fe Mössbauer spectroscopy. In contrast to the zero-field spectrum of the relaxed LS state, the numerical analysis with the Blume-Tjon relaxation model[45] of the zero-field spectrum of the trapped $HS_{TR}$ state at $T$ = 4 K now reveals two components (Fig. 6a, Supplementary Table 12 and Supplementary Fig. 48–50). The isomer shift and quadrupole splitting are similar for both components and in accord with octahedral iron(III) HS species[46–48]. The main difference between both components are the different relaxation times (in relation to the corresponding effective magnetic hyperfine fields). One component (with a volume fraction of ca. 33%) shows a resolved but still broadened magnetic hyperfine pattern with $B_{hf}$ = 51 T, which is close to the value of $|B_F|$ ≈ 55 T expected from the Fermi contact term for an iron(III) centre with an $S$ = $^5/_2$ and $L$ = 0 ($^6A_1$) electronic ground state[44]. The second component (with a volume fraction of ca. 67%) still exhibits a relaxation pattern with strong line broadening and a reduced effective magnetic hyperfine field of $B_{hf}$ = 33 T. Consistently, a two-component spectrum is also observed in supplementary measurements with an applied magnetic field of $H_{ext}$ = 5 kOe (Supplementary Fig. 44). Now, both components of the trapped HS state spectrum exhibit a well-resolved magnetic hyperfine pattern, again with similar Mössbauer parameters (Supplementary Tables 11 and 12), but different relaxation times (in relation to the corresponding effective magnetic hyperfine fields), suggesting that the observation of a two-component spectrum for the trapped HS state can be again attributed to packing effects in the solid state as

already discussed above for the relaxed LS state. If we further assume zero-field splitting for an $^6A_1$ electronic ground state and thermal population of the Kramers states (with $M_s = \pm^5/_2$, $\pm^3/_2$ and $\pm^1/_2$), the observation of longer relaxation times $\tau$ for the trapped HS state (compared to the relaxed LS state) is indicating a negative axial zero-field splitting parameter $D$. That is, with $k_B T \ll 4D$ only the lowest lying $\pm^5/_2$ Kramers state is populated, effectively suppressing fast spin-spin relaxation due to the $\Delta M_s = \pm 1$ and 0 selection rule. However, at elevated temperatures the $\pm^3/_2$ and $\pm^1/_2$ Kramers states are thermally populated, and each is contributing to the Mössbauer spectrum with an individual set of parameters ($\delta$, $\Delta E_Q$, $B_{hf}$ and fluctuation rate $\nu_c = \tau_c^{-1}$)[43]. In contrast to the LS case (S = $^1/_2$), where the population of Kramers states other than $M_s = \pm^1/_2$ is not possible, the occurrence of inequivalent $^{57}$Fe sites for the trapped HS state is therefore not unambiguously addressed. Both, a contribution of thermally populated Kramers states and/or contributions due to packing effects in the solid state, are here possible and cannot be reliably distinguished based on the available data.

To address dynamic aspects of spin state conversion in isolation, we recorded the isothermal relaxation of the magnetic moment for several hours (Fig. 7c for $50\,K \le T \le 75\,K$, Supplementary Fig. 41 for $50\,K \le T \ge 75\,K$). Rapid cooling ($\nu_\downarrow = 10\,K\,min^{-1}$) to the given temperature allowed us to establish a defined starting point, corresponding to $HS_{TR}$ ($T < 75\,K$; competing relaxation from HS cannot be avoided at higher temperatures due to the finite time demands of temperature stabilisation). As stated before, $HS_{TR}$ literally becomes kinetically inert at $T_{crit} \le 50\,K$: An extrapolated lifetime in the range of weeks indicates decay via quantum mechanical tunnelling[51,52]. Relaxation becomes progressively thermally activated at $T > 50\,K$ and shows Arrhenius behaviour (Fig. 7d).

The relaxation curves cannot be fitted to stretched exponentials, what would correspond to first-order kinetics. Rather, strong cooperativity of the HS→LS transition in compound FeB enforces self-accelerating behaviour[53–55]. To extract the activation barrier we took recourse to the kinetic model introduced by Hauser et al.[24,53–56]. From our data an activation energy $E_a = 10.3\,kJ\,mol^{-1}$ ($859\,cm^{-1}$) was determined (Arrhenius plot in Fig. 7d). While $\chi_M T$ rapidly decreases above $T_{TIESST} = 67\,K$, it is worth noting that for $T > 80\,K$ (and to a little extent at $T = 75\,K$ already), the threshold value of $\chi_M T = 0.96\,cm^3\,K\,mol^{-1}$ is not attained anymore. Then, at $T = 95\,K$, practically all iron(III) centres remain in the HS state (Supplementary Fig. 41).

The above isothermal relaxation experiment has shown that the magnetisation within the kinetic window $100\,K < T < 50\,K$ is subject to continuous decay on the time scale of several minutes to hours, but is essentially frozen at $T < 50\,K$. This opens the opportunity to convolute the time and temperature-dependent decay with a thermal ramp of variable steepness. Thus, slow temperature changes with $\nu_\downarrow < 8\,K\,min^{-1}$ do not yield the pure $HS_{TR}$ state anymore, but result in mixtures of $HS_{TR}$ and LS states, progressively enriched in the LS fraction. The resulting ratio is stationary below $T = 50\,K$. A plot of the cooling rate dependence $\nu_\downarrow$ of the $\chi_M T$ value recorded of different samples from different batches is shown in Fig. 8a (pertinent data is given in Supplementary Table 7). Efficient SCO prevails for the slowest scan rates $\nu_\downarrow$, evidently, as there is more time for the iron centres to undergo the HS→LS relaxation[22].

The most important conclusion to be drawn from this plot clearly is that a *continuum* of states can be realized (readout by the magnetometer): The stationary HS-to-LS ratio ($\gamma_{HS}$) depends on the cooling rate $\nu_\downarrow$ alone, resulting in mixtures of trapped and relaxed spin state species. Strictly speaking, cooperative SCO of the multi-inert system FeB can be tuned continuously between system states ON ($\gamma_{HS} = 1$) and OFF ($\gamma_{HS} = 0$), exhibiting a sigmoidal dependence of the quasi-stationary magnetisation with the cooling rate $\nu_\downarrow$ (at $T_{crit} = 50\,K$; cf. Figure 8b).

## Discussion

The underlying reason for compound FeB having multi-inert SCO characteristics leaves room for speculation at present. We feel that the combination of three factors is most important:

(*i*) The presence of a clear-cut cooperative connection among the spin centres. (*ii*) The conserved crystal structure which does not significantly respond to SCO, neither with respect to crystal symmetry nor with respect to cell metrics. (*iii*) The prudent choice of a counterion to govern the transfer of elastic interactions of the spin state switch.

The first argument (*i*) can be traced to the 2D H-bond network in compound FeB, which is self-regulating between 'weak' in the HS and 'moderate' in the LS state[31]. H-bonding is one of the conceptual pillars of cooperativity and imidazole moieties had proven useful in this respect[57]. Cooperative paths are likewise conveyed by imidazole units in solid FeB, but only in two lattice directions, whereas no such strong connection exists along the third lattice dimension. Hence, the rate of elastic transfer of SCO-induced strain between adjacent centres clearly must be expected to differ sharply among the lattice directions.

Absence of major structural reorganisation seems to be the key for the dynamic but highly cooperative SCO. The second argument (*ii*) thus gains some strength as conserved crystal symmetry has been rarely observed in strongly cooperative spin transitions[28,29]. This is directly linked to argument (*iii*), the role of the counterion, which

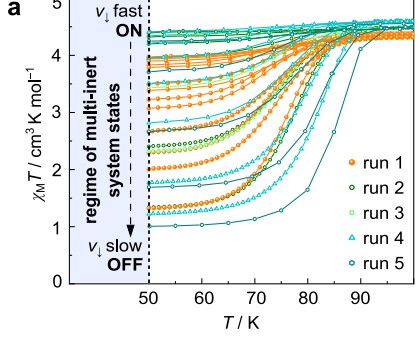
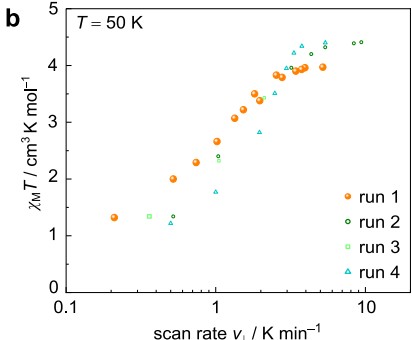

**Fig. 8 | Tuning of multi-inert states. a** Demonstration of the multi-inert SCO characteristics of compound FeB below the threshold of $T_{crit} = 50\,K$ for each of the individual series (sweep or settle mode). Solid lines are drawn for clarity and do not represent data fittings or simulations. Curves of same colour for the individual runs stem from different cooling rates (details are given in Supplementary Table 7).

**b** Logarithmic scan rate dependence of the residual paramagnetic response of compound FeB at $T = 50\,K$. Run 5 was not considered here due to the different data increments. Here, we used the *absolute* time stamps to calculate the *actual* rate $\nu$ of temperature change without any instrument lag. Source data are provided as a Source Data file.

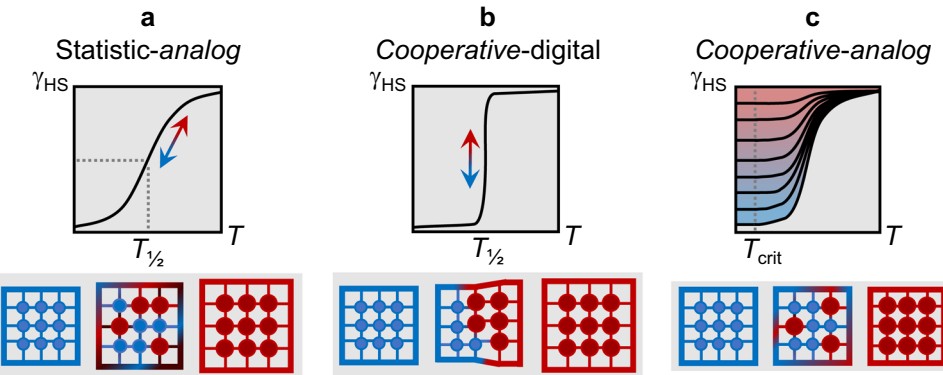

**Fig. 9 | General concepts of this work. a, b** Established pictures of cooperativity-driven spin transitions; either in Boltzmann-distributed analog-to-bistable fashion, or the digital response in an abrupt manner (see for example reference 64). **c** This work: Complementary approach of an analog system (state) response in a cooperative (multi-inert) system. Coloured circles represent the HS (red) or LS (blue) state, respectively.

appears to be crucial for the lack of metrical response. At this point, the role of the anion becomes functional for operating the multi-inert SCO.

Tetraphenylborate, as sterically demanding anion, requires an enlarged lattice, which in turn is beneficial for the larger HS molecules. Hendrickson et al. have already taken note of this circumstance and observed that $[B(Ph)_4]^-$ anions rather stabilised the HS state in a SCO system[58]. In other studies, tetraphenylborate was found to suppress the spin transition by limiting the long-range regime of intermolecular interactions[34,59]. In compound FeB, however, the anion merely softens the weak-to-moderate transition of the H-bonding network so that intermolecular interactions are retained. In a recent work, concerted atomic motion of $[B(Ph)_4]^-$ was shown to be the microscopic mechanism behind cooperative but dynamic SCO[60]. Although rotational flexibility of the anion is restricted in the present case of compound FeB, we expect the subtle change in topological shape to be critical ($\zeta$ in Fig. 3b): The interference of the anion with the mechanical elasticity of the crystal lattice keeps acceptors and donors of H-bonding in adequate proximity so that both spin states can be acquired–which eventually induces the dynamic HS→LS transition.

Taken together, it is tempting to associate the phenomenology with a matching-size effect among the lattice constituents[33,58,61]. With some caution, the occurrence of only a subtle displacement of atoms without change in crystal symmetry seems to assist in the observation of such multi-inert (or multi-metastable) transitions[15,17,24,27,62,63]. Marginal molecular motion in the solid-state is thus likely one of the driving forces to activate the fine tuning of macroscopic magnetic behaviour.

While the observation of kinetic effects on SCO is not without precedents–in fact, this phenomenon has been observed in several cases, mainly in iron(II) based systems–we feel that the prospect of multi-inert SCO systems such as compound FeB has been previously underemphasized. In particular, the importance of overlapping metastable HS regions with thermal hysteresis should be addressed. The intersection of thermodynamic and kinetic domains eventually provides access to a multi-inert regime (Figs. 8a and 9c).

For quite a number of years research has focussed on the comprehension and construction of bistable systems; while much progress has been made in microscopic modelling of the underlying cooperative paths and several systems with favourable properties (e.g., broad hysteresis around room temperature) have been identified, it is still a bistable switch with only two options to choose in a given temperature range, i.e., $\gamma_{HS}$ equals either 0 or 1 (Fig. 9a, b)[64]. By contrast, multi-inert systems such as FeB (further examples can be found in, e.g., references 15–17, 27 and 62) allow for a continuous tuning of the system state at a given temperature through variation of the thermal gradient. Below a given threshold temperature $T_{crit}$ (Fig. 9c), the system state covers the full phenomenological scale which corresponds to the set of $\gamma_{HS}(T_{crit}) = \{n \mid 0 \leq n \leq 1\}$.

The obvious paucity of kinetic effects in thermal spin crossover of iron(III) has been generally associated with only little spin-state borne inner-sphere reorganisation. Accordingly, it is no surprise that compound FeB is only the third entry to the list of TIESST-active iron(III) complexes, simply because SCO in iron(III) is significantly less structure intensive than in, e.g., iron(II). While the coordination environment of the iron(III) centre in the present case is unexceptional as is its response to the thermal spin state change, the response of the crystal lattice as a whole is unique. Lattice conservatism allows for the unique reversibility of the various spin transitions in compound FeB across many heating and cooling cycles, without any tendency for fatigue. Matching sizes of $(FeL_2)^+$ cation and $[B(Ph)_4]^-$ anion create a chessboard pattern which allows for close-to-complete resorption of the SCO-driven changes in molecular volume. Most importantly, however, cooperativity in FeB imposes kinetic effects on the spin transition that step beyond the rather common hysteresis width artifacts.

We find FeB to be multi-inert below the threshold of $T_{crit} < 50$ K. That is, at low temperature the system state can be deliberately (and remotely) selected continuously between ON and OFF, simply through selection of the proper cooling rate. Limited only by the dynamic range of the set-up, any mixture of HS and LS states becomes accessible rendering compound FeB a unidirectional, multi-inert spin state switch. We are confident that multi-inert systems can step besides the more established bistability as a complementary concept, as it offers more variable tuning of one single material: Different from cooperative SCO with bistable or digital system states, we here transform the Boltzmann-like analog response of molecular SCO from the temperature-domain to the domain of the temperature-gradient. That is, the ratio HS-to-LS in the sample is a continuous and single-valued function of the cooling rate.

## Methods
### General Remarks
All chemicals used throughout this work were commercially available (Sigma Aldrich, Fisher Chemical, TCI, abcr, Acros Organics) and were used as received. Na[B(Ph)₄] and NaOAc were dried *in vacuo* at elevated temperatures. Air-sensitive or inert preparations were carried out under Ar 5.0 atmosphere using common Schlenk tube and vacuum ($p \approx 10^{-3}$ mbar) techniques[65]. Solvents involved in preparations under inert conditions were purified by standard methods, i.e., MeOH was distilled over Mg under Ar prior to use. All other solvents were of purity grade *pro analysi* (p.a.) and used without further purification. Glassware was cleaned in a base bath (KOH in *iso*-PrOH), rinsed with H₂O and

dried at $T = 120\,°C$ overnight prior to use. 1-Benzoyl-2-(1,3-dibenzoyl-4-imidazolin-2-yl)imidazole[66,67], (1H-imidazol-2-yl)methanamine · 2 HCl[68] and 3-(methoxymethylidene)pentane-2,4-dione[69,70] were prepared following reported procedures. The amine synthesis requires a solution of HCl dissolved in iso-PrOH. Such a solution is either commercially available (5–6 M) or can be prepared by reacting $H_2SO_4$ with NaCl and passing the evolving gas (dried over $CaCl_2$) into dry iso-PrOH.

## Preparation of ligand HL

3-(Methoxymethylidene)pentane-2,4-dione (2.70 g, 19.00 mmol) is added dropwise to a suspension containing (1H-imidazol-2-yl)methanamine · 2 HCl (2.56 g, 15.06 mmol) and $Na_2CO_3$ (1.94 g, 18.3 mmol) in MeOH (100 mL). The mixture is heated for 1 h under reflux conditions, during which time a colour change into yellow is noticeable. Two methods for the work-up are possible:

(i) Once cooled to room temperature, the solution is concentrated (rotary evaporator) to obtain a semicrystalline residue. Acetone (120 mL) is added, the mixture is stirred, and excess solid is filtered off (P3 frit). Washing is not recommended. The collected solution is concentrated until a viscous, orangish oil is obtained which is left in an unsealed vessel. The waxy residue turns solid after a couple of days (about 4 d), is then crushed, separated by filtration (Büchner), washed with lots of diethyl ether, and is dried on air.

Yield: 0.93 g ($207.23\,g\,mol^{-1}$, 30%), beige solid.

(ii) Alternatively, higher yields can be obtained via Soxhlet extraction (54% based on 6.3 mmol of the amine dihydrochloride). After taking the reaction mixture to complete dryness (rotary evaporator, then in vacuo), it is crushed to a fine powder which is transferred into a cellulose-made extraction thimble (Macherey-Nagel) and the top is covered with glass wool. About 50 mL of acetone is used for each gram of crude product. The extraction time can be prolonged up to 14 days. Once the extraction is found to be complete, the volume of the collected solution is reduced by half (rotary evaporator), whereupon on cooling the ligand crystallises as fine white needles.

$^1H$ NMR (300 MHz, DMSO–$d_6$, 298 K) δ = 10.80 (1H, dt, $^3J_{HH}$ = 13.3 Hz, $^3J_{HH}$ = 6.3 Hz, NH), 10.30–8.63 (1H, br.s, NH), 8.29 (1H, d, $^3J_{HH}$ = 13.3 Hz, CH), 7.43 (2H, s, CHCH), 4.88 (2H, d, $^3J_{HH}$ = 6.3 Hz, $CH_2$), 2.29 (3H, s, $CH_3$), 2.24 (3H, s, $CH_3$) ppm.

EI–MS (pos.) m/z (%): 207 (66).

EA (calcd., found for $C_{10}H_{13}N_3O_2$, %): C (57.96, 57.33), H (6.32, 6.53), N (20.28, 20.25).

## Preparation of compound {FeL$_2$[B(Ph)$_4$]} ≡ FeB

The preparation is carried out under anhydrous conditions. The isolated compound, however, is remarkable stable as a solid under ambient conditions and can be handled without special treatment.

Ligand HL (0.201 g, 0.97 mmol), anhydrous $FeCl_3$ (0.104 g, 0.64 mmol) and NaOAc (0.094 g, 1.14 mmol) are mixed in anhydrous MeOH (40 mL) and heated for 1 h under reflux conditions. Once cooled down to room temperature, the deep red-purplish mixture is filtered (hose/cannula with glass fibre filter or P3 frit), washed with anhydrous MeOH (3 mL) and the solution is then aliquoted. This strategy is recommended for anion exchange to have a batch as back-up, since hydrolysis can occur during the procedure. To both individual batches is added Na[B(Ph)$_4$] (0.37 g, 1.08 mmol each) in one single portion, whereupon the formation of tiny red crystallites can be observed after few hours. Usually after 1 d, when the solution is only weakly coloured, the crystallites are isolated by filtration (Büchner funnel). Solid is carefully combined, washed with cold MeOH, followed by pentane, and is dried in vacuo. Care was taken to preserve the integrity of the crystallites (i.e., avoid unnecessary grinding or scratching during isolation).

Yield: 0.137 g ($787.53\,g\,mol^{-1}$, 36%), dark red needles.

HRMS (m/z): (FeL$_2$)$^+$ calcd. for $C_{20}H_{24}FeN_6O_4$, 468.12030; found 468.12000.

EA (calcd., found for $C_{44}H_{44}BFeN_6O_4$, %): C (67.11, 66.81), H (5.63, 5.35), N (10.67, 10.65).

## Single Crystal X-ray diffraction

Suitable single crystals of {FeL$_2$[B(Ph)$_4$]} ≡ FeB (obtained from bulk synthesis) were embedded in protective perfluoropolyalkylether oil and transferred to the cold nitrogen gas stream of the diffractometer. Intensity data of FeB-HS were collected at $T = 120\,K$ and of FeB-LS at $T = 65\,K$ using Mo-$K_\alpha$ radiation (λ = 0.71073 Å) on a Bruker Smart APEX 2 diffractometer using a Triumph curved graphite monochromator. The diffractometer was equipped with a N-HeliX low temperature device from Oxford Cryosystems Ltd. to maintain temperatures sub-$LN_2$. Data were corrected for Lorentz and polarisation effects; in addition, semi-empirical absorption corrections were performed on the basis of multiple scans using SADABS[71]. The structures were solved by modern dual-space algorithms (SHELXT 2014/5)[72,73] and refined by full-matrix least-squares procedures on $F^2$ using SHELXL 2018/3[74], interfaced by Olex2[75]. All non-hydrogen atoms were refined with anisotropic displacement parameters. The position of the N2-bound hydrogen atom was taken from a difference Fourier map and the positional parameters were refined in both structure determinations. All other hydrogen atoms were placed in positions of optimised geometry. The isotropic displacement parameters of all H atoms were tied to those of their corresponding carrier atoms by a factor of 1.2 or 1.5.

X-ray structure analysis of the ligand HL was performed on a Stoe StadiVari diffractometer, equipped with a graphite-monochromated Mo-$K_\alpha$ (λ = 0.71073 Å) radiation source and an Oxford Cryosystems Ltd. low-temperature unit. A suitable single crystal of ligand HL (obtained from recrystallisation in acetone) was embedded in inert perfluorinated oil (Fomblin YR-1800) and mounted on a nylon loop before collecting data at $T = 200\,K$. Data were corrected for Lorentz and polarisation effects; a spherical absorption correction was applied. The structures were solved by modern dual-space algorithms (SHELXT 2014/5)[72,73] and refined by full-matrix least-squares procedures on $F^2$ with SHELXL 2018/3[74], interfaced by WinGX[76]. All non-hydrogen atoms were refined with anisotropic displacement parameters. N-bonded hydrogen atoms were refined with free N–H distance; all other hydrogen atoms were calculated in idealized positions with fixed displacement parameters (i.e., $U_{iso}(H) = 1.5\,U_{eq}(C)$ for methyl groups and $U_{iso}(H) = 1.2\,U_{eq}(C)$ for differently connected H-atoms) during refinement.

Mercury[77] was used for structure illustrations as well as graphical output and Platon[78] for computing structural data. Octahedral distortion parameters were calculated using OctaDist[79]. Hirshfeld surface analysis[80] was achieved with CrystalExplorer[81].

## SQUID magnetometry

Magnetic measurements were conducted on SQUID magnetometer MPMS-XL5 from Quantum Design, Inc. working in the reciprocating sample option (RSO), with calibration against an internal Pd standard. Randomly distributed polycrystalline material of compound FeB was weighed into a gelatine capsule held in a plastic straw. The raw data were corrected for the diamagnetic part of the sample holder. Diamagnetic contribution $\chi_D$ from ligand and metal ion was evaluated by $M \times 0.5 \times 10^{-6}\,emu\,mol^{-1}$. We crosschecked the validity of this approximation from estimation of tabulated Pascal's constants[82,83]. Measurements were carried out at an applied field of $H = 5\,kOe$ in the range from $T = 400$-$2\,K$ in either sweep or settle mode. The MultiVu application served as graphical interface to the magnetic properties measurement system (MPMS)[84].

## $^{57}Fe$ Mössbauer spectroscopy

Polycrystalline material of compound FeB (same batch as used for SQUID magnetometry) was prepared with an area density

corresponding to ca. 0.08–0.11 mg $^{57}$Fe cm$^{-2}$ and filled in a container made of PEEK (polyether ether ketone). The measurements were conducted on commercial (WissEl, Wissenschaftliche Elektronik GmbH and Halder GmbH) transmission spectrometers with sinusoidal velocity sweep. The velocity calibration was done with an α-Fe foil at $T$ = 300 K; the minimum experimental line widths (FWHM) were <0.23 mm s$^{-1}$. The temperature-dependent measurements in zero applied magnetic fields were executed on a continuous-flow cryostat (CryoVac GmbH) with helium exchange gas, adjusted at a pressure in the sample chamber of ca. 10–50 mbar during the measurement. The temperature was controlled with a Si diode, located close to the diffusor of the cryostat, providing a temperature stability of better than ±0.1 K. The temperature of the sample was recorded with a second (but calibrated) Si diode close to the position of the sample container. The magnetic field-dependent measurements were conducted on a liquid helium bath cryostat (CryoVac GmbH) with a 5 T superconducting split-coil magnet ($H_{ext} \perp k_\gamma$). Aside from larger distances between source, sample and detector, the electronic setup and the experimental conditions are similar as in the case of the continuous-flow cryostat described above. The nominal activity of the Mössbauer sources was 50 mCi of $^{57}$Co in a Rh matrix, which was stored at ambient temperatures during the measurements. The isomer shifts were specified relative to metallic iron at $T$ = 300 K but were not corrected in terms of the second-order Doppler shift. The data analyses were carried out on basis of the stochastic Blume-Tjon relaxation model[45] using Recoil[85] and Mathematica[86] software packages.

## NMR spectroscopy

NMR spectra were recorded at room temperature with an Inova 300 spectrometer from Varian, Inc. (now Agilent Technologies, Inc.) working at $\omega_0$ = 300 MHz. The chemical shift δ in parts per million (ppm) is referenced to the residual peak of non-deuterated solvent. Signals of common impurities and solvents were assigned according to literature[87]. Coupling constants $^xJ_{n,m}$ are quoted to the nearest 0.1 Hz and are abbreviated as follows: $x$, $n$, $m$: coupling between nucleus $n$ and $m$ via $x$ chemical bonds; s: singlet; d: doublet; t: triplet; br: broad signal; or combinations thereof. Processed data were analysed using MestReNova software package.

## Elemental analysis

Analyses of elemental composition were conducted on a UNICUBE from Elementar Analysensysteme GmbH using sulfanilamide as internal standard. Samples (about 2 mg each) were weighed in a small tin boat, and the arithmetic mean of at least two measurements is reported.

## Powder X-ray diffraction

Powder X-ray diffraction of compound FeB at $T$ = 120 K was measured in transmission geometry on a Stoe StadiP diffractometer, equipped with a Ge-monochromated Cu-$K_\alpha$ radiation source and Mythen 1 K detector. Temperature control was achieved with a cryostate from Oxford Cryosystems Ltd. The polycrystalline compound FeB was ground thoroughly with an agate mortar/pestle and transferred into a glass capillary (0.5 mm ⌀).

Powder X-ray diffraction of compound FeB at $T$ = 140 and 15 K was performed using a Shimadzu XRD-7000 diffractometer with Cu-$K_\alpha$ radiation. The crystalline sample (not ground prior to measurement) was placed with a drop of EtOH on a 100 silicon zero diffraction plate. The sample plate was installed in a Janis CCS 200/202 closed cycle refrigerator from Lake Shore Cryotronics, Inc. which was coupled to the diffractometer. A Lake Shore Cryotronics, Inc. (Model 335) was used for the temperature controller.

## Mass spectrometry

Electron ionisation mass spectra were recorded with a Finnigan MAT 8500 (sector field, double focusing) from Thermo Fisher Scientific, Inc. by direct injection with a data system MASPEC II. High-resolution electrospray mass spectra were recorded with a Q Exactive (hybrid quadrupole orbitrap) from Thermo Fisher Scientific, Inc. by direct infusion. Data were analysed using the XCalibur software package.

## Optical microscopy

Photographs of polycrystalline compound FeB were taken on a Zeiss Axiovert 135 optical microscope. Scale bar was inserted using ImageJ[88].

## Data evaluation

Data evaluation was accomplished with the Origin software. Figures were prepared and edited with Microsoft Powerpoint, ChemDraw, Inkscape, Adobe Illustrator and Gimp.

## Data availability

All relevant data and other methods that are required to comprehensively follow the findings and conclusions drawn in this study are stated within the article and the Supplementary Information. This includes experimental methods, crystallographic details, spectral data (NMR, Mössbauer), and the magnetic measurements. Source data are provided with this paper. Furthermore, data can be provided from the corresponding author upon request. Crystallographic data of the structures (ligand and complex) have been deposited at the Cambridge Crystallographic Data Centre. CCDC-2210430 (for FeB-HS), and CCDC-2210431 (for FeB-LS), and CCDC-2210063 (for ligand HL) contain the supplementary crystallographic data in this paper. The data can be obtained free of charge via www.ccdc.cam.ac.uk/structures. Source data are provided with this paper.

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

## Acknowledgements

Financial support from the University of Bayreuth (UBT), the Friedrich Schiller University of Jena, the BayNAT program, and the German Research Foundation (Projects 463161096 and 509879467) is gratefully acknowledged. We thank Dr. Ulrike Lacher (Organische Chemie I, UBT) for recording mass spectra and Florian Puchtler (Anorganische Chemie I, UBT) for initial support with the powder XRD. F.W.H. thanks the Friedrich-Alexander-Universität Erlangen-Nürnberg for generous financial support. D.B. thanks Prof. Dr. Fred Jochen Litterst (Institut für Physik der Kondensierten Materie, TU Braunschweig) for providing access to the $^{57}$Fe Mössbauer spectrometers. A.D. thanks Dr. Leonhard Köhler (Organische Chemie I, UBT) for taking picturesque snapshots of the crystals.

## Author contributions

A.D., G.H., and B.W. conceptualised the work. A.D. carried out all syntheses, all SQUID measurements and routine characterisation. G.H. collected diffraction data of the ligand which was solved and refined by A.D. All Mössbauer data were collected by D.B. and M.A.C. The numerical analyses of the Mössbauer data were conducted by D.B. F.W.H. determined the crystal structures of the complex. M.A.C. performed powder X-ray diffraction measurements. A.D. wrote the first draft of the manuscript which was finalized for publication together with G.H. and B.W., supported by input and valuable contributions from all authors. B.W. oversaw the project. All authors commented on the manuscript and Supplementary Information.

## Funding

## Competing interests

The authors declare no competing interests.
