## [Peer Review File · Nature Communications]

Cooperative spin crossover leading to bistable and multiinert system states in an iron(III) complexEditorial Note: This manuscript has been previously reviewed at another journal that is not operating a transparent peer review scheme. This document only contains reviewer comments and rebuttal letters for versions considered at *Nature Communications*.

Reviewer #1 (Remarks to the Author):

In the revised manuscript the authors have adequately addressed all suggestions indicated in my original report, in particular emphasizing the novelty of the result and the ease of modulation of the spin state. The modifications of the Moessbauer description have improved the paper, as has the more detailed analysis of the role of disorder in the tetraphenylborate counterion. Overall the revised manuscript is strengthened by these revisions and publication is recommended.

Reviewer #2 (Remarks to the Author):

Overall, I deeply appreciate the efforts of the authors to revise the manuscript under consideration. My concerns regarding the novelty and a few suggested changes were adequately addressed/introduced in the response letter and in the revised version of the paper. I'm happy to recommend publication in Nat. Commun.

I have been also asked to comment on the authors' replies to the comments of reviewer #1, which I find fully satisfactory (following a careful examination). In fact, the authors "went the extra mile" to fully address the comments of all reviewers.

Reviewer #3 (Remarks to the Author):

The authors reported a Fe(III) compound exhibiting the scan rate dependent SCO behaviors. The HS state can be trapped by the rapid cooling process. The scan rate dependent SCO behaviors were then measured in details. Moreover, the Mössbauer spectra were also collected to confirm the trapped HS state. All these measurements are time-consuming and convincing. The TIESST effect has been observed in the ferrous compounds. It is rarely observed for the ferric compound. However, the TIESST effect is mainly an interesting dynamic process. The trapped HS state above 50 K will quickly go back to the LS state after three hours. I don't think the TIESST effect and the hidden hysteresis have any practical application value. Hence, this article may be suitable for more specialized journals.

1. Most of the introduction just illustrates the basic knowledge of SCO behavior.
2. The boron sublattices in Figure 4b will confuse the reader. The B atoms look like they are coordinated by the ligands.
3. The HS and LS states should be clearly shown in Table S11.
4. The magnetic curve in the settle mode should be clearly shown in Figure 7.
5. Too many data were shown in Figure 9, which can't be effectively distinguished. The scan rates should be shown in the figure or figure caption.

Reviewers' Comments:

Point-by-point replies to the individual points of the reviewers are written in green as follows: **Reply:** "....."

Reply: "We thank all reviewers for their critical and constructive comments throughout the reviewing process. In cases where we disagree with reviewer positions, we give a comment."

Reviewer #1 (Comments for the Author):

In the revised manuscript the authors have adequately addressed all suggestions indicated in my original report, in particular emphasizing the novelty of the result and the ease of modulation of the spin state. The modifications of the Moessbauer description have improved the paper, as has the more detailed analysis of the role of disorder in the tetraphenylborate counterion. Overall the revised manuscript is strengthened by these revisions and publication is recommended.

Reply: "We once again thank the reviewer for alerting us to the deficits of the original manuscript. We are happy that our changes found the reviewer's agreement."

Reviewer #2 (Remarks to the Author):

Overall, I deeply appreciate the efforts of the authors to revise the manuscript under consideration. My concerns regarding the novelty and a few suggested changes were adequately addressed/introduced in the response letter and in the revised version of the paper. I'm happy to recommend publication in Nat. Commun.

I have been also asked to comment on the authors' replies to the comments of reviewer #1, which I find fully satisfactory (following a careful examination). In fact, the authors "went the extra mile" to fully address the comments of all reviewers.

Reply: "We highly appreciate the 'warm' comments of the reviewer. We also feel that the changes made substantially improved the readability of the manuscript."

Reviewer #3 (Remarks to the Author):

The authors reported a Fe(III) compound exhibiting the scan rate dependent SCO behaviors. The HS state can be trapped by the rapid cooling process. The scan rate dependent SCO behaviors were then measured in details. Moreover, the Mössbauer spectra were also collected to confirm the trapped HS state. All these measurements are time-consuming and convincing. The TIESST effect has been observed in the ferrous compounds. It is rarely observed for the ferric compound. However, the TIESST effect is mainly an interesting dynamic process. The trapped HS state above 50 K will quickly go back to the LS state after three hours. I don't think the TIESST effect and the hidden hysteresis have any practical application value. Hence, this article may be suitable for more specialized journals.

Reply: "We regret that we could not convince the reviewer but disagree at this point (as do the other two reviewers). Clearly, relaxation below liquid nitrogen temperatures renders any technical utilization of the multi-inert manifold challenging at present. The same type of argument, however, would compromise merely the entire research on single-molecule magnets, which is an active field

still with high-level publications; *e.g.*, doi.org/10.1021/jacs.2c08568. We do not see that only the promise of ‘practical application’ warrants publication in journals with a broad readership.”

1. Most of the introduction just illustrates the basic knowledge of SCO behavior.

Reply: “Of course, the reviewer is correct in that we outline the basic features of spin crossover. We are convinced the broad and non-specialized readership of Nature Communications will benefit from a not-so-technical Introduction.”

2. The boron sublattices in Figure 4b will confuse the reader. The B atoms look like they are coordinated by the ligands.

Reply: “In the respective Figure, the boron centres appear as pink and orange spheres in the top or top-1 layer, respectively. Necessarily, the atoms in the top-1 layer are overlaid by the iron(III) centers of the top layer. We are confident that the reader will be not confused by the illustration of the boron sublattice.”

3. The HS and LS states should be clearly shown in Table S11.

Reply: “Explicit reference to the HS state and LS state is now given in the footnotes.”

4. The magnetic curve in the settle mode should be clearly shown in Figure 7.

Reply: “Thanks for this suggestion. We now give a reference in the caption.”

5. Too many data were shown in Figure 9, which can't be effectively distinguished. The scan rates should be shown in the figure or figure caption.

Reply: “Figure 9a summarizes the phenomenology of the entire set of measurements. The correspondence between the cooling rate and the remaining value of $\chi_M T$ is highlighted in Figure 9b. We thus refrain from overloading the figure with scan rate information but refer to the Supporting Information: The details of all of these sub-plots are given and discussed in the Supporting Information on pages 19-32.”